# Single-Composition White Light Emission from Dy^3+^ Doped Sr_2_CaWO_6_

**DOI:** 10.3390/ma12030431

**Published:** 2019-01-31

**Authors:** Yannan Dai, Shuai Yang, Yongkui Shan, Chun-Gang Duan, Hui Peng, Fan Yang, Qingbiao Zhao

**Affiliations:** 1School of Chemistry and Molecular Engineering, East China Normal University, Shanghai 200241, China; 51164300103@stu.ecnu.edu.cn (Y.D.); 52174300002@stu.ecnu.edu.cn (S.Y.); ykshan@chem.ecnu.edu.cn (Y.S.); 2Key Laboratory of Polar Materials and Devices, Ministry of Education, Department of Optoelectronics, East China Normal University, Shanghai 200241, China; cgduan@clpm.ecnu.edu.cn (C.-G.D.); hpeng@ee.ecnu.edu.cn (H.P.)

**Keywords:** photoluminescence, white light-emitting, double perovskite, energy transfer

## Abstract

A series of Dy^3+^ ion doped Sr_2_CaWO_6_ phosphors with double perovskite structure were synthesized by traditional high temperature solid-state method. It was found that there is significant energy transfer between Dy^3+^ and the host lattice, and the intensities of emission peaks at 449 nm (blue), 499 nm (cyan), 599 nm (orange), 670 nm (red), and 766 nm (infra-red) can be changed by adjusting the concentration of dopant amount of Dy^3+^ ion in Sr_2_CaWO_6_. The correlated color temperature of Dy^3+^ ion doped Sr_2_CaWO_6_ phosphors can be tuned by adjusting the concentration of Dy^3+^ ion. Upon optimal doping at 1.00 mol% Dy^3+^, white light with chromaticity coordinate (0.34, 0.33) was emitted under excitation at 310 nm. Thus, single composition white emission is realized in Dy^3+^ doped Sr_2_CaWO_6_.

## 1. Introduction

In recent years, phosphor-covered white light-emitting diodes (pc-WLEDs) have become more and more prevalent due to their superior performance, such as energy saving, long life time, small volume, and high brightness, compared to the conventional white light sources [1]. The commercial approach for obtaining white light-emitting diode (WLED) typically involves covering blue chips with yellow light-emitting phosphors such as Y_3_Al_5_O_12_:Ce^3+^ (YAG:Ce^3+^). Nevertheless, the lack of strong visible red-light emission makes it difficult to fabricate WLEDs with a high color-rendering index (CRI) and low correlated color temperature (CCT) [2,3,4,5,6], which limit their applications. In order to overcome these difficulties, red, green, and blue (RGB) tricolor phosphors pumped by ultra-violet (UV) LED chips (300–400 nm) are used to fabricate WLEDs with high CRI. However, it is difficult to fabricate WLED with high conversion efficiency as there is strong reabsorption of the blue-light by the green and red phosphors [7,8]. Thus, single-composition white light-emitting phosphors pumped by near ultra-violet (n-UV) light-emitting diodes are desired to fabricate WLEDs [9,10,11].

Double perovskite has the formula of A_2_BB’O_6_ [12,13]. The tungstate with double perovskite structure has important physical properties, such as magnetic [14], electrical [15], photocatalytic [16], and optical properties [17]. The luminescence properties of [WO_6_]^6−^ group in A_2_BWO_6_ has been reported [18]. As previously reported, the emission colors of [WO_6_]^6−^ group in A_2_BWO_6_ could range from blue to yellow (e.g., Ba_2_CaWO_6_, blue; Sr_3_WO_6_, green, and Ba_2_MgWO_6_, yellow), which strongly depends on the choice of A and B ions [18]. Rare earth (RE) ion doped double perovskite tungstate has been shown to have promising photoluminescence properties [19], such as Ba_2_CaWO_6_:Eu^3+^ [20], Ba_2_ZnWO_6_:Eu^3+^, Li^+^ [17], Sr_2_CaWO_6_:Eu^3+^, Na^+^ [21], and Sr_2_CaWO_6_:Sm^3+^, Na^+^ [22]. The emission peak at yellow region of Dy^3+^ ion which corresponding to the transition of ^4^F_9/2_ → ^6^H_13/2_ is greatly influenced by the coordination environment [23]. Furthermore, the ratio of emission peak intensities in blue (^4^F_9/2_ → ^6^H_15/2_) and yellow (^4^F_9/2_ → ^6^H_13/2_) region for Dy^3+^ ion is influenced by the coordination environment, and when there is an inversion center the emission in blue region is of significant intensity [24].

In the present work, Dy^3+^ was used to replace the Ca^2+^ ion in Sr_2_CaWO_6_, which is with an inversion center. Single-composition white light emitting phosphors with tunable correlated color temperature were successfully synthesized by adjusting the concentration of Dy^3+^ ions. There are some single-composition white light emission phosphors materials can be obtained at near ultra-violet(n-UV), such as Ca_9_Gd(PO_4_)_7_:Eu^2+^, Mn^2+^ [25], NaBaBO_3_:Dy^3+^, K^+^ [26], and Sr_3_Y(PO_4_)_3_:Dy^3+^ [27]. However, many materials need UV excitation under 300 nm to realize white light emission, such as Y(P,V)O_4_:Dy^3+^ [28], β-GdB_3_O_6_:Bi^3+^, Tb^3+^, Eu^3+^ [29], LaNbO_4_:Dy^3+^ [30], and LuNbO_4_:Dy^3+^ [31]. Taking LuNbO_4_:Dy^3+^ as example, the excitation peak of LuNbO_4_:Dy^3+^ phosphors centered in 261 nm. In comparison, Sr_2_CaWO_6_:Dy^3+^ phosphors can be excited under a relatively longer wavelength of 310 nm. In the present work, the photoluminescence properties of Dy^3+^ ions doped Sr_2_CaWO_6_ with different concentrations of Dy^3+^ ions and white light emission upon optimal doping are reported.

## 2. Material and Methods

The powder samples of Sr_2_Ca_(1−1.5x%)_WO_6_: x mol% Dy^3+^ (x = 0, 0.1, 0.3, 0.5, 1.0, 1.5, 2.0, 3.0) and Sr_2_Ca_0.99_WO_6_: 0.5 mol% Dy^3+^, 0.5 mol% M^+^ (M^+^ = Li^+^, Na^+^, K^+^) were synthesized by high-temperature solid-state method. The starting materials SrCO_3_ (99.9%, Sigma-Aldrich), CaCO_3_ (99.95–100.05%, Alfa), WO_3_ (99.90%, Adamas), Li_2_CO_3_ (99%, Alfa), Na_2_CO_3_ (99.8%, General-Reagent), K_2_CO_3_ (99%, Sigma-Aldrich), and Dy_2_O_3_ (99.99%, Adamas) were weighed by stoichiometric ratio. Then the starting materials were mixed and ground in an agate mortar. The mixture was transferred to a corundum crucible and preheated under 850 °C for 5 h in a box furnace. After the sample was cooled down to room temperature, the mixture was thoroughly ground, calcined at 1200 °C for 12 h, cooled to room temperature, and reground to obtain the final powder sample.

The powder XRD date of Sr_2_Ca_(1−1.5×%)_WO_6_: x mol% Dy^3+^ were collected in the range of 10° ≤ 2θ ≤ 70° using a Rigaku D/MAX 2550 diffract meter (Cu-Kα, λ = 1.54059 Å), operated at 40 kV and 40 mA. The UV–Vis absorption spectra of Sr_2_CaWO_6_ and Sr_2_CaWO_6_: doped with 1 mol% Dy^3+^ were collected with a UV–Vis spectrometer (Lambda 950, Perkin-Elmer, Waltham, MA, USA) with a polytetrafluoroethylene plate as a reference. The excitation spectra and emission spectra were collected with a photoluminescence spectrometer (FS5, Edinburgh Instruments, Livingston, UK) equipped with a 150 W xenon lamp as the excitation source. The lifetime of Sr_2_Ca_0.985_WO_6_: 1.00 mol% Dy^3+^ was measured with an FLS-980 fluorometer (Edinburgh Instruments, Livingston, UK).

## 3. Results and Discussion

### 3.1. Structural characterization

XRD patterns were collected to determine the phases of Sr_2_Ca_(1−1.5x%)_WO_6_: x mol% Dy^3+^ (x = 0, 0.1, 0.3, 0.5, 1.0, 1.5, 2.0, 3.0). Meanwhile, we also characterized the structure of Sr_2_Ca_0.99_WO_6_: 0.5% Dy^3+^, 0.5 mol% M^+^ (M^+^ = Li^+^, Na^+^, K^+^), for which Li^+^, Na^+^, and K^+^ ions were introduced as charge compensators. The ionic radius of Dy^3+^ ion in six-fold coordination is 0.912 Å, close to Ca^2+^ ion, which has an ionic radius of 1.0 Å for six-fold coordination, while the ionic radius of Sr^2+^ ion is 1.44 Å [32]. As shown in Figure 1a, the XRD patterns of Sr_2_Ca_(1−1.5x%)_WO_6_: x mol% Dy^3+^ (x = 0, 0.1, 0.3, 0.5, 1.0, 1.5, 2.0, 3.0) matched well with JPCDS card of Sr_2_CaWO_6_ (JPCDS #76-1983). As shown in Figure 1b, compared with JPCDS cards of Sr_2_CaWO_6_, the XRD patterns of Sr_2_Ca_0.99_WO_6_: 0.5 mol% Dy^3+^, 0.5 mol% M^+^ (M^+^ = Li^+^, Na^+^, or K^+^) have no obvious change. Thus, the introduction of the charge compensator has little influence on the crystal structure of Sr_2_CaWO_6_. These results indicated that the host structure of double perovskite was well preserved for both the samples with and without charge compensators.

The host compound Sr_2_CaWO_6_ is in orthorhombic system with *Pmm2* space group (a = 8.1918 Å, b = 5.7653 Å, c = 5.8491 Å, *V* = 276.24 Å^3^). In the host lattice of Sr_2_CaWO_6_, with the formula of A_2_BB’O_6_, Ca^2+^ ions and W^6+^ ions reside at B and B’ sites, respectively. Ca atoms and W atoms are coordinated by 6 O atoms (Figure 2). The cations and the coordinated oxygen ions form an octahedral structure with an inversion center. Each CaO_6_ octahedron shared its O atoms with six adjacent WO_6_ octahedron, and each WO_6_ octahedron also shared its O atoms with six adjacent CaO_6_ octahedron. Sr atoms are located at the interspace of CaO_6_ octahedron and WO_6_ octahedron and coordinated by 12 O atoms, without an inversion center [21,22].

### 3.2. UV–Vis Absorption Spectra

The UV–Vis absorption spectrum of Sr_2_CaWO_6_ is shown in Figure 3a. There is a broad absorption band in UV region. The calculated band structure and partial densities of Sr_2_CaWO_6_ and the atoms constituting Sr_2_CaWO_6_, such as strontium, calcium, tungsten, and oxygen have been reported before [22]. The strong absorption in the ultraviolet region 270–330 nm is attributed to the charge transfer from O atom to W atom. With the UV–Vis absorption spectra, the optical band gap (E_g_) of Sr_2_CaWO_6_ can be calculated with the following equation [33]:
αhν = k(hν − E_g_) ^n^(1)where α is the absorbance, h is the Planck’s constant, ν is the frequency, k is a constant, n is equal to 1/2, 2, 3/2, or 3, which is dependent on whether the transition is direct allowed, indirect allowed, direct forbidden of indirect forbidden, respectively. Wang et al reported the calculated band structure of Sr_2_CaWO_6_ and the result indicated that the Sr_2_CaWO_6_ is an indirect band gap insulator [22]. Considering the transition is indirect allowed, here n = 2. The optical band gap of Sr_2_CaWO_6_ is calculated to be 3.79 eV, while the optical band gap of Sr_2_CaWO_6_: 1.0 mol% Dy^3+^ is 3.81 eV (Figure 3b). The optical band gap of Sr_2_CaWO_6_ synthesized by sol-gel method was calculated to be 3.51 eV, which is 0.30 eV smaller than the value obtained from the present sample [22]. This suggests that the preparation conditions have appreciable influence on the optical band gap.

### 3.3. Luminescence Properties

The photoluminescence excitation spectra of Sr_2_Ca _(1−1.5x%)_ WO_6_: x mol% Dy^3+^ (x = 0, 0.1, 0.3, 0.5, 1.0, 1.5, 2.0, 3.0), which were measured at emission wavelength of 499 nm, are shown in Figure 4a. As shown in Figure 4a, there is a broad excitation band in the region of 270–330 nm. The calculated band structure and total densities of states of Sr_2_CaWO_6_ near the Fermi energy level have been reported [22]. The broad excitation band centered at 310 nm was attributed to the charge transfer from O^2−^ to W^6+^ ions. The doping concentrations of Dy^3+^ ion have significant influence on the excitation band of the host lattice. With the increase of doping concentration, the excitation intensity at 310 nm decreased. The concentration dependent excitation intensity of phosphors at 310 nm is shown in Figure 4b.

The photoluminescence emission spectra of Dy^3+^ doped Sr_2_CaWO_6_ phosphors were collected under excitation at 310 nm. Figure 5a shows the emission spectra of Sr_2_Ca_0.99_WO_6_: 0.5 mol% Dy^3+^, 0.5 mol% M^+^ (M^+^ = Li^+^, Na^+^ or K^+^). The introduction of charge compensation ions had no significant influence on the emission of Dy^3+^ ion doped Sr_2_CaWO_6_ phosphors. A broad emission band centered at 449 nm, which is due to self-trapped luminescent recombination in [WO_6_]^6−^ octahedral [19,34]. The emission peaks at 499 nm (^4^F_9/2_ → ^6^H_15/2_), 599 nm (^4^F_9/2_ → ^6^H_13/2_), 670 nm (^4^F_9/2_ → ^6^H_11/2_) and 766 nm (^4^F_9/2_ → ^6^H_9/2_) are attributed to f–f transitions of Dy^3+^ ions. As shown in Figure 5a, the emission intensity of Sr_2_Ca_0.99_WO_6_: 0.5 mol% Dy^3+^, 0.5 mol% M^+^ decreased in different degrees with introducing charge compensation ions, which indicated that the defects caused by the introduction of charge compensation ions cause more energy loss. The emission spectra of Sr_2_Ca_(1−1.5x%)_WO_6_: x mol% Dy^3+^ (x = 0, 0.1, 0.3, 0.5, 1.0, 1.5, 2.0, 3.0) is shown in Figure 5b. The concentration-dependent emission intensity variations at 449 nm, 499 nm, 599 nm and 670 nm are shown in Figure 5c.

The electric dipole transition of ^4^F_9/2_ → ^6^H_13/2_ emission of Dy^3+^ ion is sensitive to surrounding environment [24]. If there is no inversion center, the ^4^F_9/2_ → ^6^H_13/2_ emission of Dy^3+^ ions will be strong. Otherwise, the ^4^F_9/2_ → ^6^H_13/2_ emission of Dy^3+^ ion will be weak. However, the transition of ^4^F_9/2_ → ^6^H_15/2_ is not as sensitive to coordinate surroundings [24,35]. Therefore, the symmetry of the environment in which Dy^3+^ ions are located can be judged by comparing the relative intensity of ^4^F_9/2_ → ^6^H_13/2_ and ^4^F_9/2_ → ^6^H_15/2_ transition [36]. In most fluorescent materials, with Dy^3+^ ions in the asymmetric position the transition intensity of ^4^F_9/2_ → ^6^H_13/2_ is much stronger than that of ^4^F_9/2_ → ^6^H_15/2_ (such as SrMoO_4_:Dy^3+^ [37], LuNbO_4_:Dy^3+^ [31], and Sr_2_ZnWO_6_:Dy^3+^ [19]). With Dy^3+^ ion located at a position with high symmetry, the transition intensity of ^4^F_9/2_ → ^6^H_13/2_ is almost the same as that of ^4^F_9/2_ → ^6^H_15/2_, or even weaker than that of ^4^F_9/2_ → ^6^H_15/2_ (such as Ba_3_La_2−x_(BO_3_)_4_:xDy^3+^ [38], Ba_2_Ca_(1−x)_WO_6_:xDy^3+^ [20], Sr_3_Sc_1−x_(PO_4_)_3_:xDy^3+^ [39]). As can be seen from the emission spectra of Dy^3+^ ion doped Sr_2_CaWO_6_, the emission intensity of ^4^F_9/2_ → ^6^H_15/2_ at 499 nm is similar to ^4^F_9/2_ → ^6^H_13/2_ centered at 599 nm, which indicates that Dy^3+^ replaces the position with high symmetry.

As shown in Figure 6, the excitation peaks located at 352 nm, 366 nm, and 455 nm are attributed to the f-f transition absorptions of Dy^3+^ ions. The excitation spectra of Dy^3+^ ions have significant overlap with the emission spectra of Sr_2_CaWO_6_, and there is energy radiation transfer from Sr_2_CaWO_6_ host lattice (donors) to Dy^3+^ ions (acceptors) [40]. Hence, the emission peak intensity at 449 nm decreased with the increase of the concentration of Dy^3+^ ion, while the emission peak intensities at 499 nm, 599 nm, 670 nm, and 766 nm increased with higher doping concentration of Dy^3+^ ion when x ≤ 1.0. However, when x >1.0, the emission peak intensities at 499 nm, 599 nm, 670 nm, and 766 nm decreased with the increase of the concentration of Dy^3+^ ion. The emission peak-intensities at 449 nm, 499 nm, 599 nm, and 670 nm changed with concentration of Dy^3+^ ion, and single-composition WLED phosphors with tunable correlated color temperature were successfully generated through adjusting the concentration of Dy^3+^ ion.

The critical distance (*R*_c_) between Dy^3+^ ions were calculated by the concentration quenching method. The critical transfer distance (*R*_c_) was calculated with the following formula (Equation (2)) which was proposed by Blasse [41]:(2)$$Rc\approx 2{\left[\frac{3V}{4\pi x_{c}N}\right]}^{\frac{1}{3}}$$

In this equation, *V* is the volume of crystallographic unit cell, x_c_ is the critical concentration, and N is the lattice site number in a unit cell which can be replaced by sensitizers. In Sr_2_CaWO_6_, *V* = 276.24 Å^3^, *N* = 2, and *x*_c_ = 0.01, *R*_c_ of Dy^3+^ ion is calculated to be 29.8 Å. In general, for the energy transfer process, the exchange interaction and multipole interaction are the two mechanisms that can play important roles [40]. Exchange interactions take place over a distance shorter than 5 Å, while multipole interaction can occur at a distance as large as 30 Å [42,43]. The critical distance of Dy^3+^ ion in Sr_2_CaWO_6_ equal to 29.8 Å, which is much larger than 5 Å. Therefore, the energy transfer process belongs to multipole interaction instead of exchange interaction.

Van Uitert [40] has pointed out that when non-radiative losses are attributed to multipolar transfer, the strength of multipolar interaction can be determined from the change in the emission intensity of activator. The relation between the emission intensity of each activator and the concentration of each activator can be expressed by Equation (3) [44,45]:
(3)$$\mathrm{lg}\frac{I}{x}=-\frac{s}{3}\mathrm{lg}x+A$$where *I* is the integral emission intensity of ^6^F_5/2_ → ^4^H_13/2_, *x* is the corresponding doping concentration, A is a constant which independent on the dopant concentration, and s is dependent on the interaction process. The value of s can be 6, 8, and 10, corresponding to electric dipole–dipole, electric dipole–quadrupole or electric quadrupole–quadrupole interaction, respectively. When *s* equal to 3, the energy transfer among nearest-neighbor ions plays a major role in quenching. As shown in Figure 7, the slope was calculated to be −0.87, thus s is most approximate 3. Therefore, the energy transfer process is most likely caused by energy transfer among nearest-neighbor ions.

Figure 8 shows the decay curve of Sr_2_CaWO_6_: 1.00 mol% Dy^3+^ phosphors excited at 310 nm and monitored at 499 nm. The decay curve fits well with the following single-exponential Equation (4):
*I*(*t*) = *A exp* (−*t*/*τ*)(4)where *I*(*t*) is the emission intensity at time t and A is a constant. Thus, the lifetime value of *τ* is calculated to be 0.48 ns. The reported lifetime value of Sr_1.99_CaWO_6_:0.01Dy^3+^ is 127 μs [34]. Therefore, synthesis methods and doping sites appear to have significant influence on fluorescence lifetime.

The proposed energy transfer mechanism of Sr_2_CaWO_6_:Dy^3+^ phosphors are shown in Figure 9. In Sr_2_CaWO_6_, electrons at valance band top were excited under 310 nm ultraviolet irradiation and transferred to conduction band, which is mainly attributed to the charge transfer from O atoms to W atoms. Electrons at conduction band returned to conduction band bottom through non-radiative transition. When electrons at conduction band bottom transfer to the top of valance band, energy is released by radiative transition. Thus, there is a broad blue light emission band in Sr_2_CaWO_6_. In Sr_2_Ca_(1−1.5x%)_WO_6_: x mol% Dy^3+^ phosphors, besides the excitation of charge transfer from O atoms to W atoms, Dy^3+^ ions are also excited by the ultraviolet under 310 nm. Dy^3+^ ions can be excited to energy levels higher than ^4^F_9/2_ by visible light from 350 nm until 450 nm. Electrons at high energy levels return to ^4^F_9/2_ configuration through non-radiative transition, then release to ^6^H_J_ (J = 11/2, 13/2, 15/2) configuration through radiative transition. It is well known that Dy^3+^ ions have matched energy level pairs which produce strong cross relaxation and lead to concentration quenching. Therefore, it can be inferred that besides the energy transfer among nearest-neighbor ion, the concentration quenching is also related to cross relaxation of Dy^3+^ ions (^4^F_9/2_:^6^H_15/2_ → ^6^H_9/2_ + ^6^F_11/2_:^6^F_3/2_ and ^4^F_9/2_:^6^H_15/2_ → ^6^F_5/2_:^6^H_7/2_ + ^6^F_9/2_, where ^6^H_9/2_ + ^6^F_11/2_ and ^6^H_7/2_ + ^6^F_9/2_ mean the energy level of ^6^H_9/2_ and ^6^H_7/2_ are very close to those of ^6^F_11/2_, and ^6^F_9/2_, respectively) [45].

### 3.4. Commission International de I’Eclairage (CIE) chromaticity diagram

As shown in Table 1, with the increase of doping concentration, the CIE coordinates of Sr_2_Ca_(1-1.5x%)_WO_6_: x mol% Dy^3+^ phosphors can be adjusted from the blue region (0.18, 0.16) to white (0.34, 0.33) (x = 1), which is very close to the coordinate of standard white light (0.33, 0.33). As the doping concentration continues to increase, the CIE coordinates gradually shift to the yellow region (0.37, 0.36). By adjusting the concentration of Dy^3+^ ion, a series of phosphors with different CIE coordinates were successfully obtained and white light from a single host was successfully obtained (Figure 10).

Chromaticity coordinates of Sr_2_Ca_(1−1.5x%)_WO_6_: x mol% Dy^3+^ (x = 0, 0.1, 0.3, 0.5, 1.0, 2.0, 3.0) and Sr_2_Ca_0.99_WO_6_: 0.5 mol% Dy^3+^, 0.5 mol% M^+^ (M^+^ = Li^+^, Na^+^ or K^+^) were compared at various Dy^3+^ ion concentration. It can be seen that the CIE coordinates of the samples with and without charge compensation ions are very close (Figure 10). Therefore, the introduction of charge compensation ion does not significantly affect the chromaticity coordinates.

## 4. Conclusions

In summary, phosphors with Dy^3+^ doped on the Ca site of Sr_2_CaWO_6_ were prepared by high temperature solid state method, and they can be excited under 310 nm ultraviolet. The host compound, Sr_2_CaWO_6_ emits blue light centered at 449 nm with the color coordinate of (0.18, 0.16) under ultraviolet excitation at 310 nm. The intensity of emission peaks under 310 nm excitation can be tuned by adjusting the concentration of Dy^3+^ ion. White light emission with CIE coordinate (0.34, 0.33) was successfully generated in Sr_2_CaWO_6_:Dy^3+^ phosphors at the doping level of 1 mol% on the Ca site. Considering the overlap between the emission spectra of host lattice and the excitation spectra of Dy^3+^ ions, it is expected that there is efficient energy transfer from host lattice to Dy^3+^ ions.

## Figures and Tables

**Figure 1 materials-12-00431-f001:**
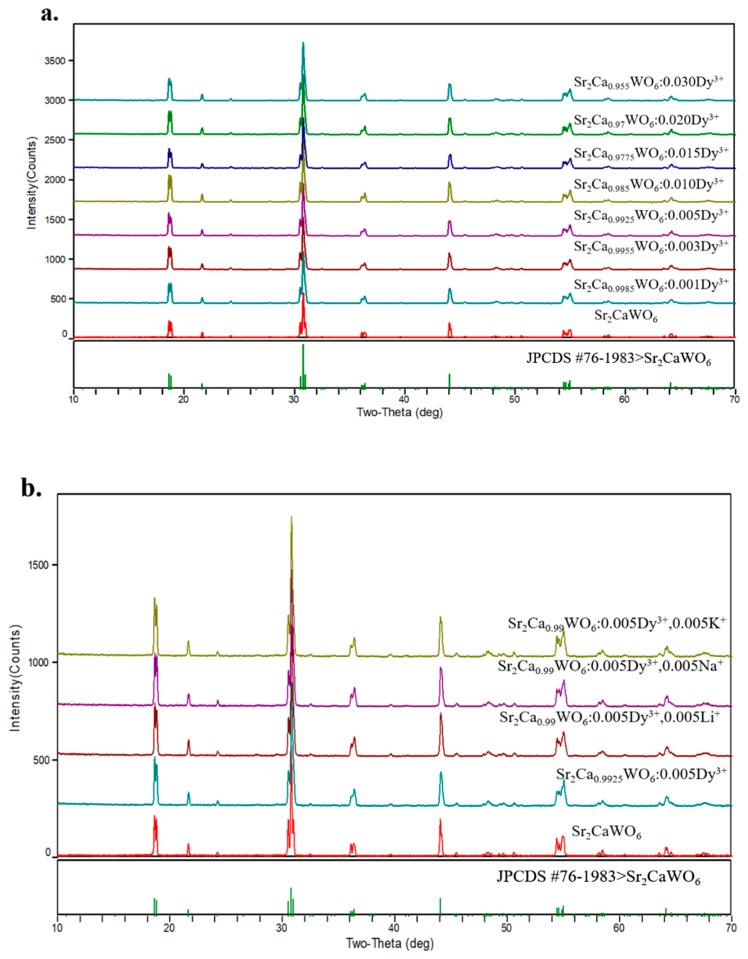
(**a**). The X-ray diffraction patterns of Sr_2_Ca_(1−1.5x%)_WO_6_: x mol% Dy^3+^ (x = 0, 0.1, 0.3, 0.5, 1.0, 1.5, 2.0, 3.0); (**b**). The X-ray diffraction patterns of Sr_2_CaWO_6_, Sr_2_Ca_0.9925_WO_6_: 0.5 mol% Dy^3+^ and Sr_2_Ca_0.99_WO_6_: 0.5 mol% Dy^3+^, 0.5 mol% M^+^ (M^+^ = Li^+^, Na^+^, or K^+^).

**Figure 2 materials-12-00431-f002:**
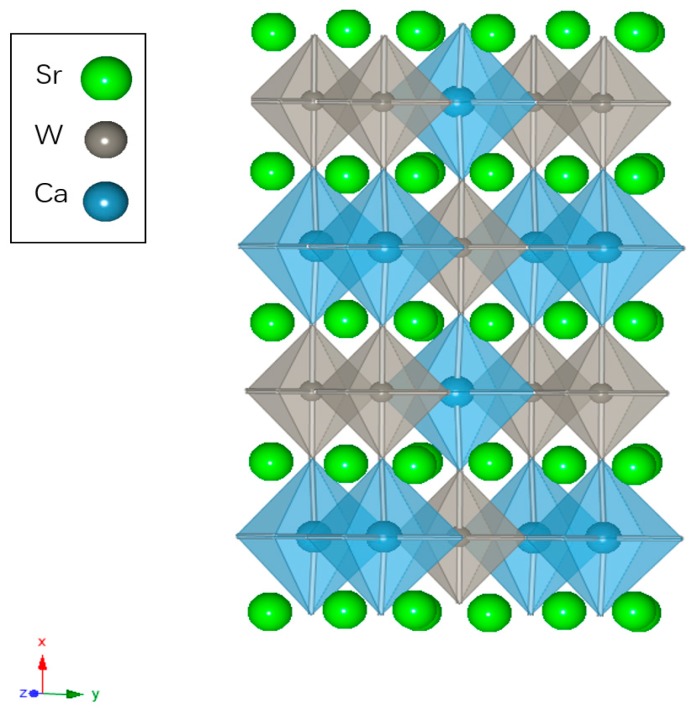
The crystal structure of Sr_2_CaWO_6_ in which Sr atoms are marked with green ball, gray WO_6_ octahedrons and cerulean CaO_6_ octahedron are shown to represent coordination of W atoms and Ca atoms.

**Figure 3 materials-12-00431-f003:**
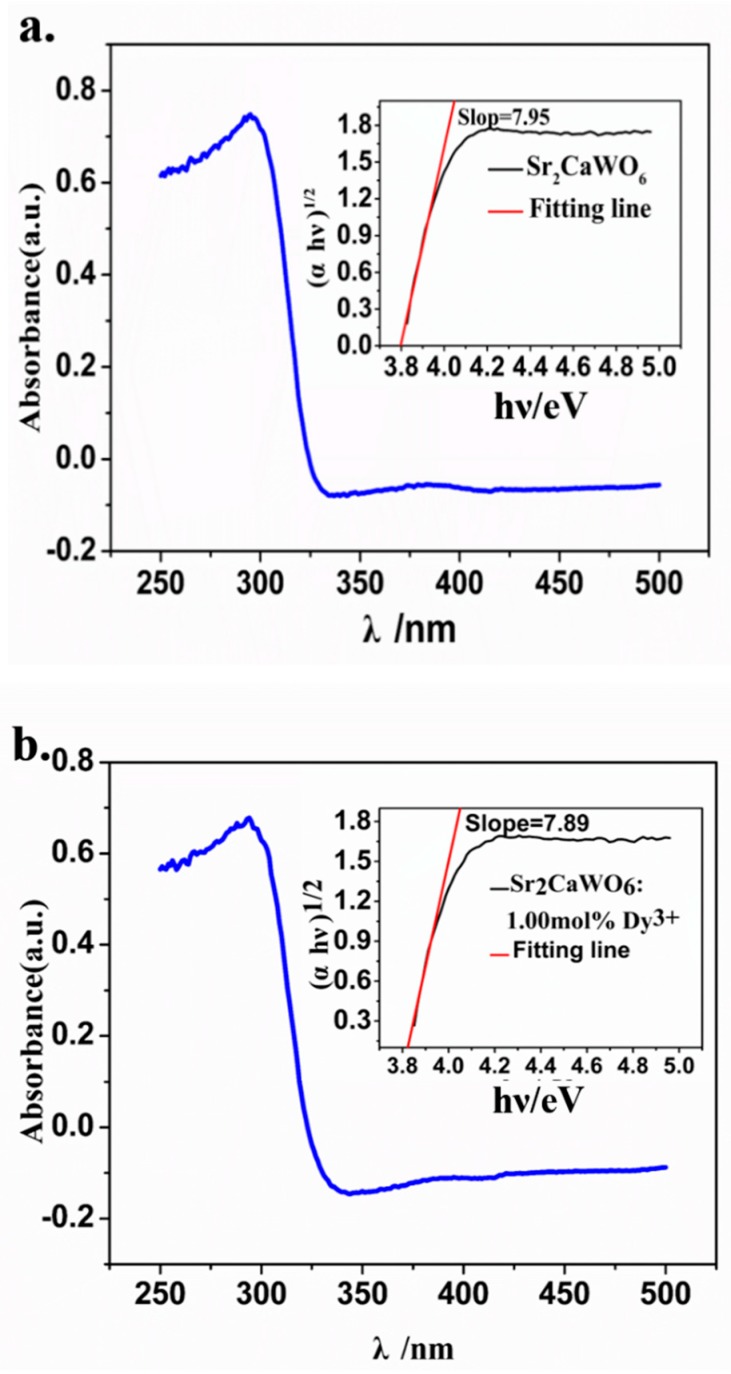
The UV–Vis absorption spectra of Sr_2_CaWO_6_ (**a**) and Sr_2_CaWO_6_: 1.0 mol% Dy^3+^ (**b**), the insert shows variation of (αhν)^1/2^ under different photon energy.

**Figure 4 materials-12-00431-f004:**
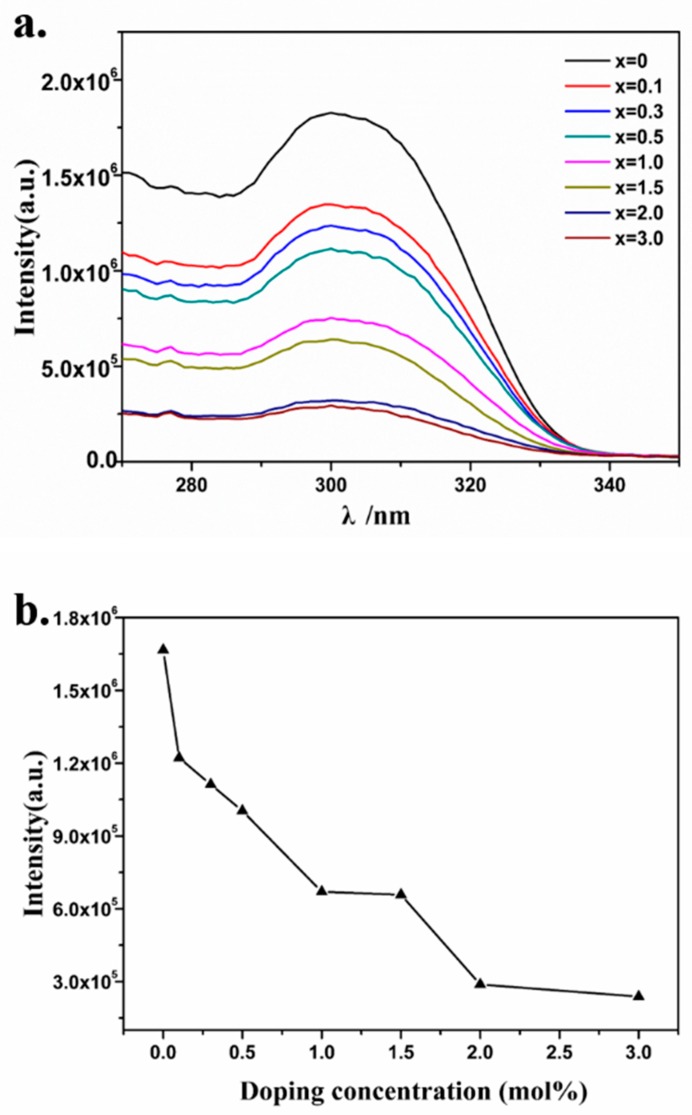
(**a**) The photoluminescence excitation spectra (λ_em_ = 499 nm) of Sr_2_Ca_(1−1.5x%)_WO_6_: x mol% Dy^3+^ (x = 0, 0.1, 0.3, 0.5, 1.0, 1.5, 2.0, 3.0). (**b**) The concentration-dependent excitation intensity variation of phosphors at 310 nm.

**Figure 5 materials-12-00431-f005:**
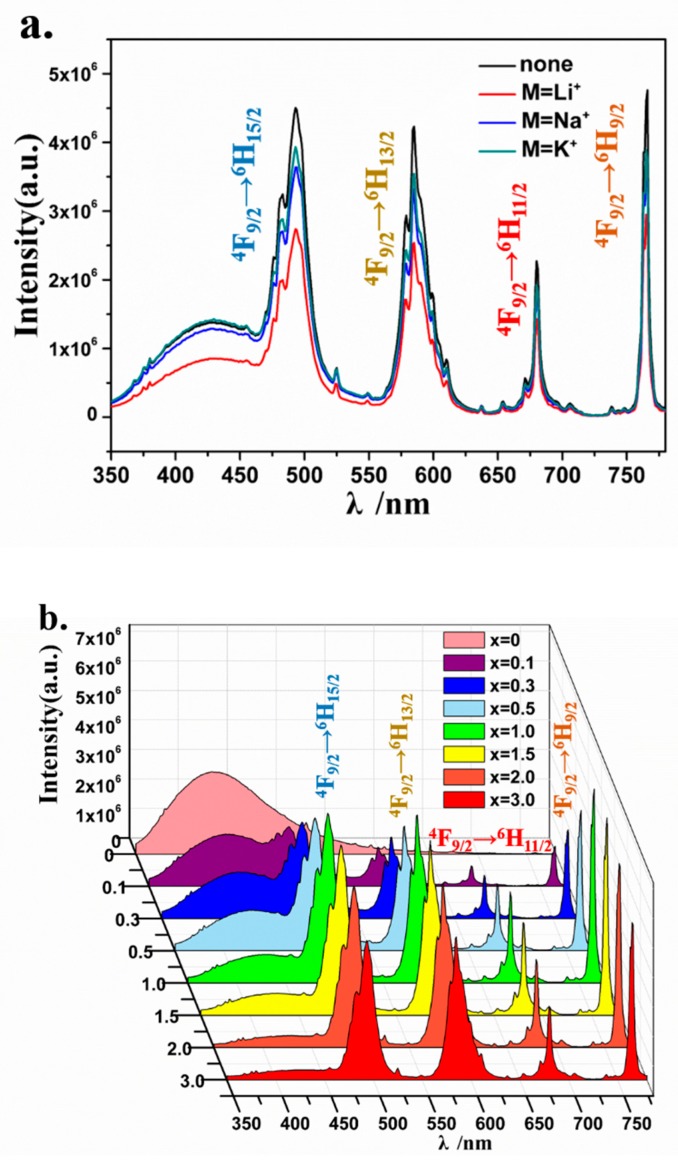
(**a**) Emission spectrum of Sr_2_Ca_0.99_WO_6_: 0.5 mol% Dy^3+^, 0.5 mol% M^+^ (M^+^ = Li^+^, Na^+^ or K^+^) under 310 nm excitation; (**b**) Emission spectrum of Sr_2_Ca_(1−1.5x%)_WO_6_: x mol% Dy^3+^ (x = 0, 0.1, 0.3, 0.5, 1.0, 1.5, 2.0, 3.0) under 310 nm excitation. (**c**) The concentration-dependent emission intensity variations at 449 nm, 499 nm, 599 nm and 670 nm respectively.

**Figure 6 materials-12-00431-f006:**
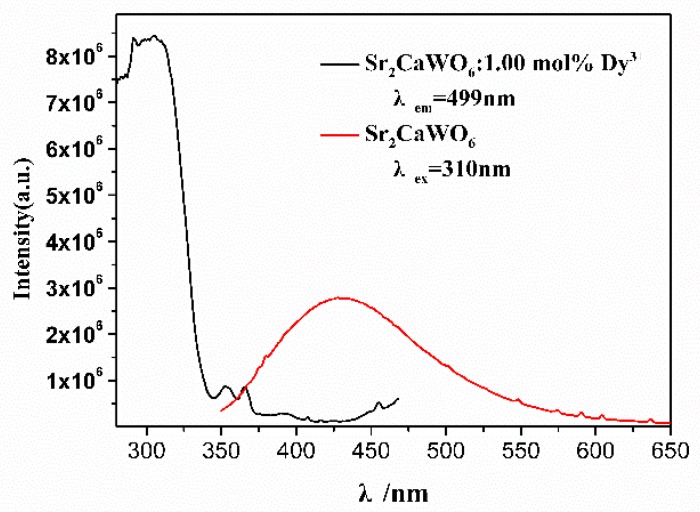
The excitation spectrum of Sr_2_Ca_0.985_WO_6_: 1 mol% Dy^3+^ (λ_em_ = 499 nm) is shown in black, and the emission spectrum of Sr_2_CaWO_6_ under 310 nm excitation is shown in red.

**Figure 7 materials-12-00431-f007:**
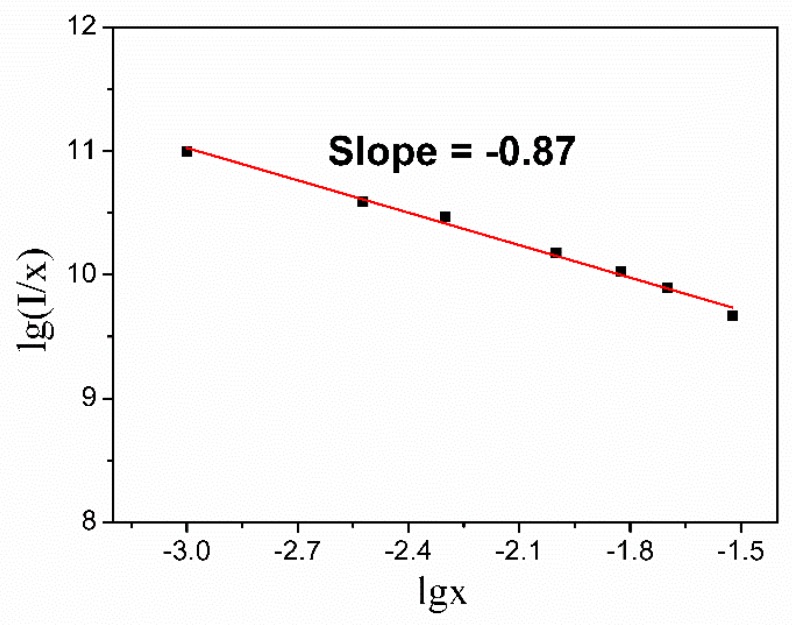
The function relation between lg (*I*/*x*) and lg x.

**Figure 8 materials-12-00431-f008:**
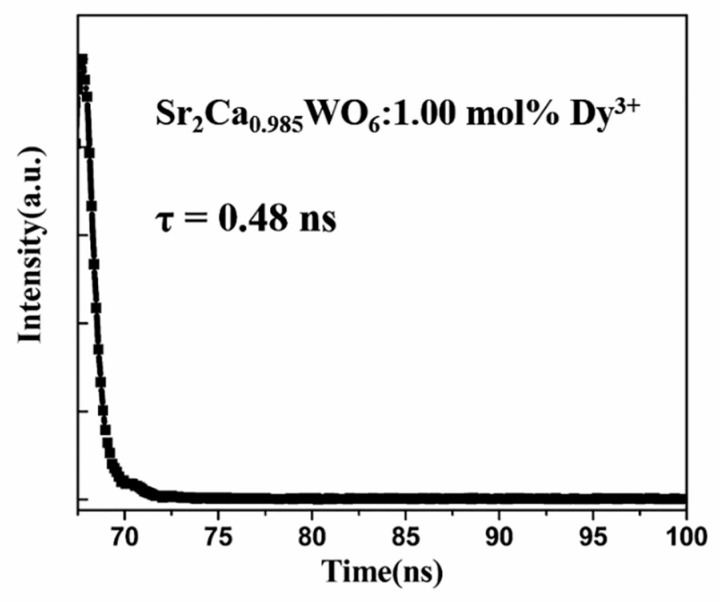
Decay curve of Sr_2_CaWO_6_: 1.00 mol% Dy^3+^ at room temperature with λ_ex_ = 310 nm and λ_em_ = 499 nm.

**Figure 9 materials-12-00431-f009:**
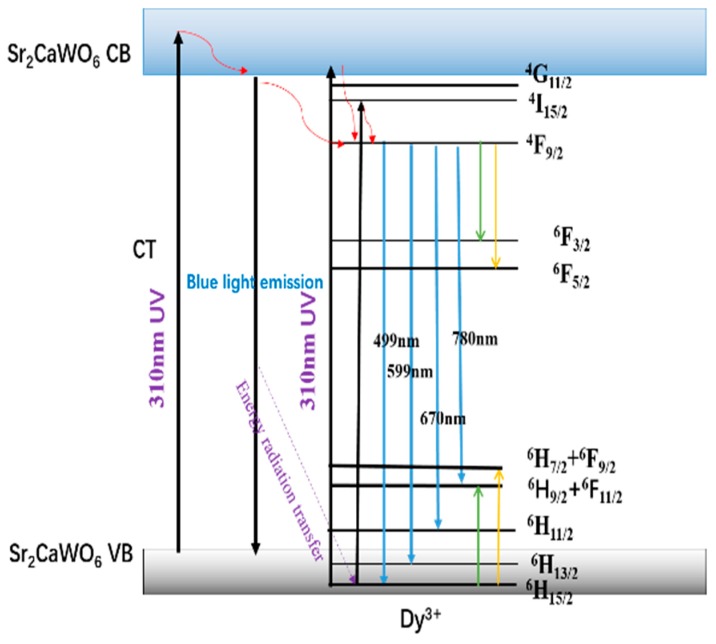
Schematic illustration of the energy transfer mechanism for Sr_2_CaWO_6_:Dy^3+^. The red solid curve with an arrow means the vibrational relaxation of excited Dy^3+^ ion in Sr_2_CaWO_6_. The green solid line with an arrow and the yellow one means the cross relaxation ^4^F_9/2_:^6^H_15/2_ → ^6^H_9/2_ + ^6^F_11/2_:^6^F_3/2_ and ^4^F_9/2_:^6^H_15/2_ → ^6^F_5/2_:^6^H_7/2_ + ^6^F_9/2_ of Dy^3+^ ions, respectively.

**Figure 10 materials-12-00431-f010:**
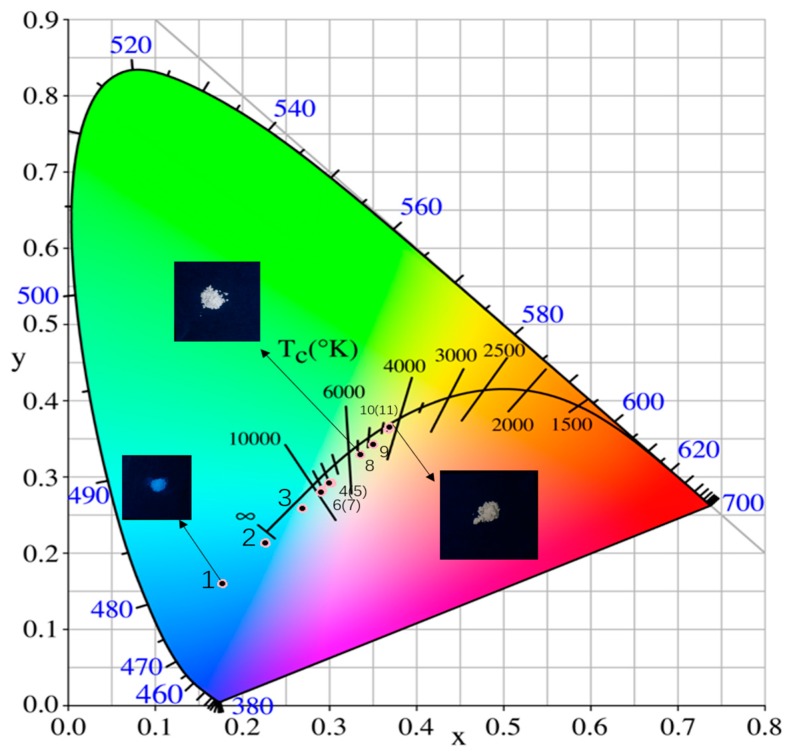
CIE 1931 color coordinates of Sr_2_Ca_(1-1.5x%)_WO_6_: x mol% Dy^3+^ (x = 0, 0.1, 0.3, 0.5, 1.0, 1.5, 2.0, 3.0) and Sr_2_Ca_0.99_WO_6_: 0.5 mol% Dy^3+^, 0.5 mol% M^+^ (M^+^ = Li^+^, Na^+^ or K^+^).

**Table 1 materials-12-00431-t001:** Commission International de I’Eclairage (CIE) coordinates of different Dy^3+^ ion concentration of Sr_2_CaWO_6_.

Number	Phosphors	CIE Coordinates (x,y)
1	Sr_2_CaWO_6_	(0.18, 0.16)
2	Sr_2_Ca_0.9985_WO_6_:0.001Dy^3+^	(0.23, 0.21)
3	Sr_2_Ca_0.9955_WO_6_:0.003Dy^3+^	(0.27, 0.26)
4	Sr_2_Ca_0.9925_WO_6_:0.005Dy^3+^	(0.30, 0.29)
5	Sr_2_Ca_0.99_WO_6_:0.005Dy^3+^,0.005Li^+^	(0.30, 0.29)
6	Sr_2_Ca_0.99_WO_6_:0.005Dy^3+^,0.005Na^+^	(0.29, 0.28)
7	Sr_2_Ca_0.99_WO_6_:0.005Dy^3+^,0.005K^+^	(0.29, 0.28)
8	Sr_2_Ca_0.985_WO_6_:0.010Dy^3+^	(0.34, 0.33)
9	Sr_2_Ca_0.9775_WO_6_:0.015Dy^3+^	(0.35, 0.34)
10	Sr_2_Ca_0.97_WO_6_:0.020Dy^3+^	(0.37, 0.36)
11	Sr_2_Ca_0.955_WO_6_:0.030Dy^3+^	(0.37, 0.36)

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
