# Peer review of "Single-Composition White Light Emission from Dy3+ Doped Sr2CaWO6"

_materials, 2019, doi:10.3390/ma12030431_

Reviewer 1 Report

    The manuscript entitled Single-composition white light emission from Dy3+ doped Sr2CaWO6 presents interesting results. The obtained phosphors are characterized by different methods. Being interesting, the manuscript can be further improved in order to increase its scientific value. Thus, the Authors are asked:

    To improve the quality of the figures: 6 and 7.

    To discuss the possible maximum content of Dy3+ ions which can be introduced into presented phosphors.

    To discuss the luminescence lifetimes of the Dy3+ in obtained materials. Please provide the experimental results and discussion with the already published results.

    To explain if the presented concentration-dependent emission/excitation intensity variations are presented based on integrated intensity? It seems that it's just the presentation of the raw data. 

Author Response

    The manuscript entitled Single-composition white light emission from Dy3+ doped Sr2CaWO6 presents interesting results. The obtained phosphors are characterized by different methods. Being interesting, the manuscript can be further improved in order to increase its scientific value. Thus, the Authors are asked:

     1. To improve the quality of the figures: 6 and 7.

    Response: The quality of the figures: 6 and 7 are improved.

    2. To discuss the possible maximum content of Dy3+ ions which can be introduced into presented phosphors.

    Response: Thank the reviewer for asking this question. According to your suggestion, we have synthesized some Sr2Ca(1-1.5x%)WO6:x mol%Dy3+ phosphors with higher doping content. Even if x equals to 40 mol%, the crystal structure can be well maintained. However, since the concentration quenching occurs at a rather low doping concentration, the discussion of the effect of high concentration doping ions on the crystal structure was not put into the manuscript. 

    3. To discuss the luminescence lifetimes of the Dy3+ in obtained materials. Please provide the experimental results and discussion with the already published results.

    Response: The decay curve of Sr2CaWO6: 1.00 mol% Dy3+ phosphors excited at 310 nm and monitored at 499 nm, the value of lifetime is calculated to be 0.48 ns. 

    4. To explain if the presented concentration-dependent emission/excitation intensity variations are presented based on integrated intensity? It seems that it's just the presentation of the raw data. 

    Response: Thank the reviewer for asking this question. Raw data were used for the intensity. In Fig. 4 (a) and Fig. 5 (b), since the shape of the peak in the corresponding excitation and emission spectra does not change, raw data are representative and were used to indicate the trend of intensity with concentration.

Reviewer 2 Report

see attached

Author Response

    Review of “Single-composition white light emission from Dy3+ doped Sr2CaWO6,” by Yannan Dai, Shuai Yang, Yongkui Shan, Chun-Gang Duan, Hui Peng, Fan Yang, and Qingbiao Zhao (manuscript: materials-422091) 

    The authors present evidence for synthesis of luminescent white light emitting phosphors. The information is relevant to the phosphor research community, and may be of interest to other thin film luminescence researchers. The experiments and results are mostly well laid out, and seem to support the arguments presented by the authors. The manuscript needs some relatively heavy English editing (i.e., should clips in line 27 be “chips”?). Also, many of the figures are difficult to read, and should be larger (specifically: Figure 1, Figure 3, Figure 6, Figure 8 and Figure 9. It couldn’t hurt to have the other ones slightly larger, too). The title and abstract need to have the correct chemical subscripts and superscripts. Scientifically, I have a couple of criticisms:

    1. Dy3+ is not a drop-in replacement for Ca2+, as the authors acknowledge in Table 1, but not in section 3.1 (or elsewhere). They should really write the formula as: Sr2Ca(1-1.5x%)WO6: mol% Dy3+, which supports the charge neutrality of 2 Dy3+ ions replacing 3 Ca2+ ions in the lattice during doping.The XRD does not change much with the doping, as the authors point out, so the overall crystal structure does not appear to distort under these low doping conditions, but, nonetheless, the stoichiometry should be stated correctly. 

    Response: We thank the reviewer for pointing this issue out. Now all these formulas were rewritten as Sr2Ca(1-1.5x%)WO6: x mol% Dy3+

    2. The authors invoke cross-relaxation for the dysprosium ions in their energy level diagram (Figure 8) and the surrounding text, but they should point out that they have not presented experimental evidence for this, but that it may help explain the quenching observed as the dopant level increases.

    Response: Dy3+ ions have matched energy level pairs which produce strong cross relaxation and lead to concentration quenching. In the new manuscript it was pointed out that cross relaxation is likely the reason for concentration quenching, and the diagram was not meant to be a representation of experimental results.

    3. Also in Table 1, the authors present the CCT column for the different powders, but do not discuss it anywhere in the text. Either introduce it in the discussion and talk about its relevance, or get rid of it. Also, the temperatures are much too specific: is it really 14252.09 K for 0.3 mol% Dy3+? Or should the correct significant figures be more like 14252 or 14250 K?

    Response: Thank the reviewer for pointing this out. Since it is not a significant part of this paper, the CCT column has been deleted.

    These issues must be addressed before the manuscript can be published

Reviewer 3 Report

    The system was studied [Xiaoyu Song et al. Crystal structure and magnetic-dipole emissions of Sr2CaWO6: RE3+ (RE=Dy, Sm and Eu) phosphors Journal of Alloys and Compounds 739 (2018) 660; Synthesis and photoluminescence properties of microcrystalline Sr2ZnWO6:RE3+ (RE = Eu, Dy, Sm and Pr) phosphors].

In the present case the authors used solid state reaction which is simple but not very controllable (a “harsh” one) because it requires high temperatures annealing that rise the question of non-stoichiometry due to the elements evaporation and synthesis related defects.

Other observations:

    1. Regarding the: “Considering the transition is indirect allowed, here n = 2.” The authors have to explain/discuss this assumption…see Xiaoyu Song et al. Crystal structure and magnetic-dipole emissions of Sr2CaWO6: RE3+ (RE=Dy, Sm and Eu) phosphors Journal of Alloys and Compounds 739 (2018) 660

    2. The assignment of the broad blue luminescence band from Fig. 5a is questionable….the blue emission is likely to be due to self-trapped luminescent recombination in WO6 octahedral complex [Xiaoyu Song et al. Journal of Alloys and Compounds 739 (2018) 660] and not CT as the authors claim, being similar to those of alkaline-earth tungstates with the perovskite structure [see K.V. Dabre et al. J. Alloy. Comp. 617 (2014) 129; J.C. Sczancoski et al. J. Colloid Interface Sci. 330 (2009) 227

    3. It would be useful to mention about the influence on the luminescence spectra of the 0.5mol% M+ (M+ =Li+, Na+, or K+) charge compensators by comparison to the un-codoped Sr2CaWO6:Dy3+ [X. Song et al. / Journal of Alloys and Compounds 739 (2018) 660 and K.V. Dabre et al. Synthesis and photoluminescence properties of microcrystalline Sr2ZnWO6:RE3+ (RE = Eu, Dy, Sm and Pr) phosphors Journal of Alloys and Compounds 617 (2014) 129–134] where RE3+ is incorporated in low symmetry sites

    4. It was claimed that Sr2CaWO6:Dy3+ with Dy3+ doping the Ca2+ site is a promising single composition white light emitting phosphor but without any measurement (quantum yield as in ref [33]) or at least a comparative estimation.

    5. Some of the notations and insets in the figures are unclear, hardly to see/read.

Author Response

    The system was studied [Xiaoyu Song et al. Crystal structure and magnetic-dipole emissions of Sr2CaWO6: RE3+ (RE=Dy, Sm and Eu) phosphors Journal of Alloys and Compounds 739 (2018) 660; Synthesis and photoluminescence properties of microcrystalline Sr2ZnWO6:RE3+ (RE = Eu, Dy, Sm and Pr) phosphors].

    Response: The Sr2CaWO6 is interesting in that both the Sr and the Ca site can be doped by rare earth elements. In the report of Song et al, Sr site of Sr2CaWO6 was doped by Dy and no white color was found. In the present study, the Sr2CaWO6 was doped on the Ca site, and while light emission was realized.

    In the present case the authors used solid state reaction which is simple but not very controllable (a “harsh” one) because it requires high temperatures annealing that rise the question of non-stoichiometry due to the elements evaporation and synthesis related defects.

    Response: The elements we used are not very volatile, and the XRD pattern shows that the crystal structure does not have significant change. This indicate that the amount of defects is minor.

 Other observations:

    1. Regarding the: “Considering the transition is indirect allowed, here n = 2.” The authors have to explain/discuss this assumption…see Xiaoyu Song et al. Crystal structure and magnetic-dipole emissions of Sr2CaWO6: RE3+ (RE=Dy, Sm and Eu) phosphors Journal of Alloys and Compounds 739 (2018) 660

    Response: The energy band structure of Sr2CaWO6 (space group: Pmm2) has been calculated by previous report, and pointed out that the Sr2CaWO6 is an indirect band gap insulator. For indirect band gap insulator, n = 2 with the equation we used. On this basis,we calculatted the optical bandgap of the as-prepared Sr2CaWO6.

    2. The assignment of the broad blue luminescence band from Fig. 5a is questionable…the blue emission is likely to be due to self-trapped luminescent recombination in WO6 octahedral complex [Xiaoyu Song et al. Journal of Alloys and Compounds 739 (2018) 660] and not CT as the authors claim, being similar to those of alkaline-earth tungstates with the perovskite structure [see K.V. Dabre et al. J. Alloy. Comp. 617 (2014) 129; J.C. Sczancoski et al. J. Colloid Interface Sci. 330 (2009) 227

    Response: We thank the reviewer for pointing this out. In the revised manuscript, the broad blue luminescence band is attributed to the self-trapped luminescent recombination in WO6 octahedral complex.

     3. It would be useful to mention about the influence on the luminescence spectra of the 0.5mol% M+ (M+ =Li+, Na+, or K+) charge compensators by comparison to the un-codoped Sr2CaWO6:Dy3+ [X. Song et al. / Journal of Alloys and Compounds 739 (2018) 660 and K.V. Dabre et al. Synthesis and photoluminescence properties of microcrystalline Sr2ZnWO6:RE3+ (RE = Eu, Dy, Sm and Pr) phosphors Journal of Alloys and Compounds 617 (2014) 129–134] where RE3+ is incorporated in low symmetry sites

    Response: Because of the introduction of alkali metal ions, the blue emission and the characteristic emission of Dy3+ are weakened, so alkali metal ions are not introduced in the following discussion. Although some rare earth ions have been reported to enter the low symmetric position, in fluorescent materials with various doping sites possibly provided by alkaline earth metals, different doping positions can be achieved by controlling the proportion of raw materials. ( Jayakiruba S , et al. Excitation-dependent local symmetry reversal in single host lattice Ba2A(BO3)2:Eu3+ [A = Mg and Ca] phosphors with tunable emission colours[J]. Physical Chemistry Chemical Physics, 2017, 19(4):17383-17395. ). In this paper, Dy replaced the position of Ca2+ ion by changing the amount of CaCO3 and keeping the amount of SrCO3 constant. The emission intensity of 4F9/2→6H15/2 at 499 nm is very similar to 4F9/2→6H13/2 centered at 599 nm, which indicates that Dy3+ replaces the position with high symmetry.

    4. It was claimed that Sr2CaWO6:Dy3+ with Dy3+ doping the Ca2+ site is a promising single composition white light emitting phosphor but without any measurement (quantum yield as in ref [33]) or at least a comparative estimation.

    Response: The reason why we think Sr2CaWO6: Dy3 + with Dy3+ is of certain significance is that the blue-emitting Sr2CaWO6 matrix can be synthesized by a simple synthesis method. On this basis, the yellow-emitting Dy3+ ions are introduced to achieve near-standard white emission by adjusting the concentration of Dy3+ ions. We have revised the statement into the following in the abstract: Thus, single composition white emission is realized in Dy3+ doped Sr2CaWO6.

    5. Some of the notations and insets in the figures are unclear, hardly to see/read.

    Response: We thank the reviewer for pointing this out. Pictures with better readability were given.
